# Correction: Lagrois et al. Low-to-Mid-Frequency Monopole Source Levels of Underwater Noise from Small Recreational Vessels in the St. Lawrence Estuary Beluga Critical Habitat. *Sensors* 2023, *23*, 1674

**DOI:** 10.3390/s23229143

**Published:** 2023-11-13

**Authors:** Dominic Lagrois, Camille Kowalski, Jean-François Sénécal, Cristiane C. A. Martins, Clément Chion

**Affiliations:** 1Département des Sciences Naturelles, Université du Québec en Outaouais, Ripon, QC J0V 1V0, Canada; 2Parc Marin du Saguenay-Saint-Laurent, Tadoussac, QC G0T 2A0, Canada


**Text Correction**


There was an error in the original publication [1], **that phrase is meant to indicate that there is an absence of auditory stresses**.

**A correction has been made to*****5. Discussion***, ***Paragraph 2***:

“…would have caused any specific auditory stresses…” should be “…would have caused no specific auditory stresses…”

There was an error in the original publication [1]. Conclusion #2 states that “the received noise levels [⋯] mostly fell below the St. Lawrence Estuary beluga hearing audiogram hence suggesting that they caused limited acoustic disturbance at CPA distances of a few hundreds meters”. Following the standard peer-review process, acceptation, and publication of the manuscript, it was pointed out to us that the comparison between the RLs recorded at the hydrophone and the beluga’s audiogram’s detection thresholds was not done appropriately, hence calling into question the validity of the original paper’s Conclusion #2.


**Henceforth, a correction has been made to Conclusion #2:**
2.At least 31.1% of the recorded targets (14 out of 45 events) in Anse-Saint-Étienne have shown received noise levels in excess of the St. Lawrence Estuary beluga hearing audiogram, hence suggesting evidence for acoustic disturbance at CPA distances of a few hundreds meters. In those specific cases, both the beluga’s communication and echolocation bands have increased risks of auditory masking during short-to-intermediate range interactions (<600 m) (see Appendix D).



**Newly added “Appendix D. Proof”, as follow,**



**Appendix D. Proof**


To address any form of misinterpretation, we closely followed the method described by Gervaise and collaborators [53, *§* IV-B]. Figure A1 shows, in light gray, the RLs recorded by the hydrophone (in units of dB Hz${}^{-1}$) during CPA passage for all 45 events listed in this work’s Table 1. Critical bands of twelfth-octave are generally suited to the beluga’s hearing sensitivity [73,74]. In the frequency domain, the subtraction of critical ratios [75] from the RL twelfth-octave bands leads to a spectrum that can now be directly compared to published hearing audiograms. Positive differences (referred to as excesses), shown in Figure A1 as circle-chained lines, correspond the amount (in dB) of a signal at a given frequency that must be emitted above the audiogram to be detected.

The audiograms in Figure A1 are a mixture of auditory evoked potentials and behavioural hearing thresholds. The main objective of this work was to contribute/add to the sample of known MSLs for motorised recreational crafts (see, e.g., [56]); it is beyond its scope to determine which audiogram is better representative of the beluga’s hearing thresholds and why. In Table A2, masking was judged probable if at least four out six audiograms suggest so with a maximal signal excess required for detection above 4.8 dB (highlighted in coral), i.e., the nominal uncertainty on a single acoustic measurement [76]. All in all, 14 out of 45 events, i.e., 31.1% of our sample in Anse-Saint-Étienne, support evidence for masking at the position of the hydrophone, in addition to 5 contested events.

**Figure A1 sensors-23-09143-f0A1:**
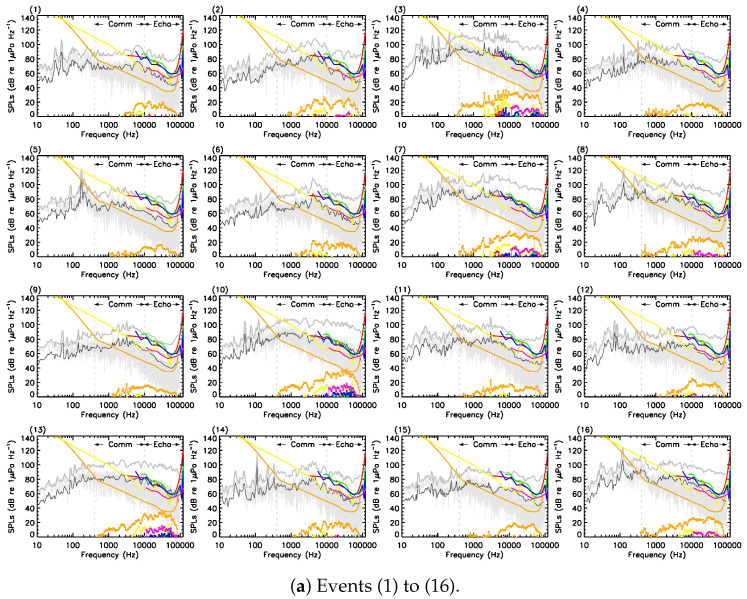
Acoustic disturbance in Anse-Saint-Étienne. Each panel (and label) corresponds to a specific event in [1]’s Table 1. At CPA, RL levels (in units of dB Hz${}^{-1}$) recorded at the hydrophone are shown in light gray. The same signal expressed in twelfth-octave bands (in units of dB) is provided in gray. Subtraction of the frequency-dependent critical ratios [75] from the twelfth-octave bands yields the black curve. Six (6) audiograms for the beluga whale are provided: red [77], orange [5], yellow [45], green [78], blue [79], and magenta [80]. Subtraction of each audiogram from the black curve gives the color-corresponding circle-chained line (referred as excesses in the text) in the lower portion of each panel. Beluga’s communication and echolocation bands are represented by the vertical dashed gray lines.

**Table A2 sensors-23-09143-t0A2:** Maximal signal excess required for detection and corresponding frequency.

Event	Castellote et al. [77]		Erbe et al. [5]		Finneran et al. [45]		Klishin et al. [78]		Mooney et al. [79]		Sysueva et al. [80]		Masking?
	**Frequency**	**Excess**	**Frequency**	**Excess**	**Frequency**	**Excess**	**Frequency**	**Excess**	**Frequency**	**Excess**	**Frequency**	**Excess**	
	**(kHz)**	**(dB)**	**(kHz)**	**(dB)**	**(kHz)**	**(dB)**	**(kHz)**	**(dB)**	**(kHz)**	**(dB)**	**(kHz)**	**(dB)**	
1	−	−	13.6	21	7.2	8	−	−	−	−	13.6	2	✗
2	4.5	3	8.5	27	8.5	12	−	−	−	−	40.6	6	?
3	4.8	14	8.1	37	6.8	26	57.5	7	8.1	12	15.2	15	✓
4	−	−	20.3	19	7.2	4	−	−	−	−	−	−	✗
5	−	−	28.7	16	−	−	−	−	−	−	−	−	✗
6	4.0	3	4.0	22	6.8	7	−	−	−	−	−	−	✗
7	4.5	8	10.2	35	7.2	17	45.6	6	15.2	5	15.2	14	✓
8	3.0	2	10.8	28	51.2	11	51.2	1	−	−	11.4	8	?
9	−	−	3.2	18	6.8	3	−	−	−	−	−	−	✗
10	36.2	8	36.2	38	51.2	20	54.2	10	36.2	8	36.2	18	✓
11	3.0	5	3.0	23	6.8	6	−	−	−	−	−	−	?
12	−	−	10.8	25	8.5	7	−	−	−	−	12.1	4	✗
13	54.2	7	30.4	36	51.2	18	54.2	11	54.2	7	30.4	15	✓
14	−	−	8.0	21	8.0	8	−	−	−	−	21.5	1	✗
15	−	−	13.6	19	48.3	4	−	−	−	−	−	−	✗
16	−	−	30.4	28	51.2	11	51.2	1	−	−	11.4	7	?
17	−	−	28.7	9	−	−	−	−	−	−	−	−	✗
18	4.0	10	9.1	31	7.2	17	45.6	3	7.6	3	12.8	11	✓
19	−	−	7.6	23	7.6	12	−	−	−	−	19.2	2	✗
20	−	−	13.6	18	9.1	3	−	−	−	−	−	−	✗
21	−	−	8.1	23	7.6	12	−	−	−	−	−	−	✗
22	−	−	10.2	16	−	−	−	−	−	−	−	−	✗
23	4.3	1	4.3	21	7.2	5	−	−	−	−	15.2	1	✗
24	4.0	18	4.0	38	4.0	21	−	−	−	−	12.1	9	✓
25	51.2	4	51.2	29	51.2	18	51.2	8	51.2	4	51.2	10	✓
26	4.3	1	4.2	21	7.2	9	−	−	−	−	−	−	✗
27	4.0	1	10.8	24	7.6	8	−	−	−	−	11.4	4	✗
28	−	−	27.1	22	7.2	6	−	−	−	−	27.1	1	✗
29	−	−	17.1	21	8.1	4	−	−	−	−	17.1	2	✗
30	−	−	38.4	27	48.3	9	48.3	1	−	−	38.4	8	?
31	−	−	3.0	12	−	−	−	−	−	−	−	−	✗
32	−	−	4.3	18	6.8	2	−	−	−	−	−	−	✗
33	3.0	1	12.1	20	7.2	7	−	−	−	−	12.1	0	✗
34	3.8	6	38.4	27	7.2	14	48.3	1	−	−	38.4	8	✓
35	−	−	30.4	10	−	−	−	−	−	−	−	−	✗
36	38.4	19	10.2	50	9.1	33	43.1	20	9.1	20	13.6	29	✓
37	4.0	15	30.4	38	51.2	23	54.2	13	57.5	10	38.4	17	✓
38	3.0	18	3.4	37	7.6	20	51.2	6	8.1	8	13.5	14	✓
39	−	−	38.4	9	−	−	−	−	−	−	−	−	✗
40	3.2	8	18.1	31	9.1	15	51.2	5	18.1	4	18.1	13	✓
41	3.4	13	3.4	31	3.4	13	−	−	16.1	1	15.2	10	✓
42	−	−	10.2	15	−	−	−	−	−	−	−	−	✗
43	3.4	6	8.1	32	8.1	19	48.3	3	8.1	7	11.4	8	✓
44	−	−	30.4	18	7.2	3	−	−	−	−	−	−	✗
45	3.8	1	40.6	22	7.2	6	−	−	−	−	15.2	3	✗


**References**



**A correction has been made to References (added 8 more references based on the original 72 references):**
73. Erbe, C.; Farmer, D.M. A software model to estimate zones of impact on marine mammals around anthropogenic noise. *J. Acoust. Soc. Am.* **2000**, *108*, 1327–1331.74. Erbe, C.; Farmer, D.M. Zones of impact around icebreakers affecting beluga whales in the Beaufort Sea. *J. Acoust. Soc. Am.* **2000**, *108*, 1332–1340.75. Johnson, C.S.; McManus, M.W.; Skaar, D. Masked tonal hearing thresholds in the beluga whale. *J. Acoust. Soc. Am.* **1989**, *85*, 2651–2654.76. Sponagle, N. Variability of Ship Noise Measurements. In *Defense Research Establishment Atlantic*; Technical Report; Defence Research Establishment Atlantic: Dartmouth, NS, Canada, 1988.77. Castellote, M.; Mooney, T.A.; Quakenbush, L.; Hobbs, R.; Goertz, C.; Gaglione, E. Baseline hearing abilities and variability in wild beluga whales *(Delphinapterus leucas)*. *J. Exp. Biol.* **2014**, *217*, 1682–1691.78. Klishin, V.O.; Popov, V.V.; Supin, A.Y. Hearing capabilities of a beluga whale, *Delphinapterus leucas*. *Aquat. Mamm.* **2000**, *26*, 212–228.79. Mooney, T.A.; Castellote, M.; Quakenbush, L.; Hobbs, R.; Gaglione, E.; Goertz, C. Variation in hearing within a wild population of beluga whales *(Delphinapterus leucas)*. *J. Exp. Biol.* **2018**, *221*, jeb171959.80. Sysueva, E.V.; Nechaev, D.I.; Popov, V.V.; Supin, A.Y. Electrophysiological audiograms in seven beluga whales (Delphinapterus leucas) from the Okhotsk Sea population. In Proceedings of the Meetings on Acoustics, Acoustical Society of America, Minneapolis, MN, USA, 7–11 May 2018; Volume 33, p. 010001.


## References

[B1-sensors-23-09143] Lagrois D., Kowalski C., Sénécal J.F., Martins C.C.A., Chion C. (2023). Low-to-Mid-Frequency Monopole Source Levels of Underwater Noise from Small Recreational Vessels in the St. Lawrence Estuary Beluga Critical Habitat. Sensors.

