# Peer review of "Correction: Lagrois et al. Low-to-Mid-Frequency Monopole Source Levels of Underwater Noise from Small Recreational Vessels in the St. Lawrence Estuary Beluga Critical Habitat. Sensors 2023, 23, 1674"

_sensors, 2023, doi:10.3390/s23229143_

Round 1
Reviewer 1 Report
This paper present data and analysis from the St. Lawrence estuary recreational boat traffic and its potential influence on Beluga communication/hearing.
The protocol and analysis are sounds and well explained. My only comment before publication would be to try to improve the readability of some of the figures. In particular, figure 2 lacks contrast, which makes it hard to read without zooming in on each panel, which in turns makes comparing the events harder.
Another comment would be to add the 120dB line in figure 6 since it is the current regulatory level. This would help make the discussion in the text clearer.
Overall, this is a good paper. The above suggestions simply aim at making it more readable.
Author Response
We thank to reviewer for his/her comments:
Q: My only comment before publication would be to try to improve the readability of some of the figures. In particular, figure 2 lacks contrast, which makes it hard to read without zooming in on each panel, which in turns makes comparing the events harder.
R: Figures 2 and 4 (previously 3) were reprocessed to display only the events that are mentionned in the text.
Q: Another comment would be to add the 120dB line in figure 6 since it is the current regulatory level. This would help make the discussion in the text clearer.
R: This is addressed in Figure 7 (previously 6).
Reviewer 2 Report
The paper “Low-to-mid frequency monopole source levels of underwater noise from small recreational vessels in the St. Lawrence Estuary Beluga critical habitat” explores the radiated noise emitted by small recreational vessels that thrive in many coastal waters, such as in the St. Lawrence Estuary Beluga population’s summer habitat, with tests carried out during summers of 2021 and 2022.
The amount of data gathered by the authors is remarkable; nevertheless, some adjustments are fundamental and can increase the value of the manuscript.
- Appendices are very useful to understand data analysis and evaluation. However, it is preferable to introduce important aspect of the theory used for the data processing inside the main test. I suggest removing most of the text inside the appendices B and C and introduce it inside sections of interest. Furthermore, the text lacks a theoretical structure: please consider the possibility to introduce more detail about the theoretical background of your experiments.
- Line 43: “..an endangered species protected under 42 the [25]. In this case, it is better to write the name of the cited report, and then cite it as you properly did.
- Line 67-69: Please consider to introduce a citation here which help to find online the datasheet of the used hydrophone to the reader.
- Figure 2 and 3: I understand that authors tried to compress acquired data in these figures, but the resolution is too low, and a printed document would be impossible to read. Please increase the resolution of each picture. Indeed, it is hard to follow the discussion in lines 204-210.
Author Response
We thank the reviewer for his/her comments:
Q: Appendices are very useful to understand data analysis and evaluation. However, it is preferable to introduce important aspect of the theory used for the data processing inside the main test. I suggest removing most of the text inside the appendices B and C and introduce it inside sections of interest.
R: A large portion of Appendix C has been moved to Section 3.4. What remains in Appendices B and C are very simple methods for frequency integration and the use of the SONAR equations. These are very basic concepts in acoustics and we find it appropriate to leave them in appendix to facilitate the lecture.
Q: Furthermore, the text lacks a theoretical structure: please consider the possibility to introduce more detail about the theoretical background of your experiments.
R: A more theoretical paragraph has been added at lines 160 to 169.
Q: Line 43: “..an endangered species protected under 42 the [25]. In this case, it is better to write the name of the cited report, and then cite it as you properly did.
R: The name of the cited report as been added to main text.
Q: Line 67-69: Please consider to introduce a citation here which help to find online the datasheet of the used hydrophone to the reader.
R: Citation has been added at line 69.
Q: Figure 2 and 3: I understand that authors tried to compress acquired data in these figures, but the resolution is too low, and a printed document would be impossible to read. Please increase the resolution of each picture. Indeed, it is hard to follow the discussion in lines 204-210.
R: Both figures have been reprocessed and now show only (bigger) panels that are mentionned in the discussion.